# Evaluating the Efficacy of a Peripheral Nerve Simulator-Guided Brachial Plexus Block in Rabbits Undergoing Orthopaedic Surgery Compared to Systemic Analgesia

**DOI:** 10.3390/vetsci11050213

**Published:** 2024-05-13

**Authors:** Sophie A. Mead, Matthew J. Allen, Sara Ahmed Hassouna Elsayed, Claudia S. Gittel

**Affiliations:** 1Queen’s Veterinary School Hospital, University of Cambridge, Cambridge CB3 0ES, UK; claudia.gittel@rossdales.com; 2Southfields Veterinary Specialists, Basildon SS14 3AP, UK; 3Surgical Discovery Centre, University of Cambridge, Cambridge CB3 0ES, UK; 4Faculty of Veterinary Medicine, Alexandria University, Alexandria 5424041, Egypt; 5Department of Veterinary Medicine, University of Cambridge, Cambridge CB3 0ES, UK; 6Rossdales Equine Hospital, Cotton End Rd, Newmarket CB8 7NN, UK

**Keywords:** locoregional anaesthesia, multimodal analgesia, lidocaine CRI, faecal output, nerve stimulator, pain score, activity, behaviour, video processing

## Abstract

**Simple Summary:**

Local anaesthetic nerve blocks can be used to provide pain relief during and after surgery. While specific local anaesthetic techniques are commonplace in humans and some species, it is a relatively new field of research in rabbits. Rabbits are prey species that hide pain well but may express changes in behaviour, food intake, and production of faeces as a result of pain, making them challenging to study. This study aimed to investigate a specific local anaesthetic technique in rabbits undergoing orthopaedic surgery on a front leg. Its effectiveness was investigated by comparing the requirement for extra pain relief during and after surgery and comparing changes in food intake, faeces production, and behaviour after surgery. Both remote filming and direct observation were used. The rabbits who received the block required no additional pain relief during surgery, whereas every rabbit who received intravenous pain relief did require additional pain relief. However, after surgery, the severity of pain and the requirement for extra pain relief were the same, and there was no difference in behaviour between the groups. In conclusion, this local anaesthetic nerve block was easy to administer and provided effective pain relief during surgery, reducing the need for additional drug therapy.

**Abstract:**

Locoregional anaesthetic techniques are invaluable for providing multimodal analgesia for painful surgical procedures. This prospective, randomised study describes a nerve stimulator-guided brachial plexus blockade (BPB) in rabbits undergoing orthopaedic surgery in comparison to systemic lidocaine. Premedication was provided with intramuscular (IM) medetomidine, fentanyl, and midazolam. Anaesthesia was induced (propofol IV) and maintained with isoflurane. Nine rabbits received a lidocaine BPB (2%; 0.3 mL kg^−1^), and eight received a lidocaine constant rate infusion (CRI) (2 mg kg^−1^ IV, followed by 100 µg kg^−1^ min^−1^). Rescue analgesia was provided with fentanyl IV. Carprofen was administered at the end of the surgery. Postoperative pain was determined using the Rabbit Grimace Scale (RGS) and a composite pain scale. Buprenorphine was administered according to the pain score for two hours after extubation. Rabbits were filmed during the first two hours to measure distance travelled and behaviours. Food intake and faeces output were compared. Every rabbit in CRI required intraoperative rescue analgesia compared to none in BPB. However, rabbits in both groups had similar pain scores, and there was no difference in the administration of postoperative analgesia. There were no significant differences in food intake or faeces production over 18 h, and no significant differences in distance travelled or behaviours examined during the first two hours. BPB seems superior for intraoperative analgesia. Postoperatively, both groups were comparable.

## 1. Introduction

In accordance with the orthopaedic surgery literature, surgical procedures are often associated with moderate to severe pain, meaning that suitable intraoperative and postoperative analgesia regimens are of paramount importance [1,2]. Pain and stress in the postoperative period are associated with several physiological changes, including sympathetic nervous system activation and endocrine changes [3,4,5]. These lead to protein catabolism and hyperglycaemia, causing impaired wound healing, weight loss, and infection [3,4]. Therefore, adequate perioperative pain management is mandatory for painful procedures and requires regular pain assessment. With rabbits commonly used as an animal model in experimental orthopaedic surgeries, accurate pain assessment remains particularly challenging due to their tendency to suppress pain behaviours in light of being prey animals. This further emphasises the importance of recognising and managing pain using balanced anaesthesia protocols.

The use of locoregional anaesthesia techniques for various surgical procedures, now commonplace in dogs and cats, has been reported in other species, including sheep [6], calves [7], goats [8,9], and pigs [10,11]. Local nerve blocks may be performed blind or guided by electrical nerve stimulation or ultrasound. Electrical nerve stimulation facilitates more precise localisation of the nerves than using anatomical landmarks alone. Ultrasound-guided local blocks have been associated with increased block success, faster onset of effect, increased duration, reduced risk of vascular puncture, and reduced volume of drug required [12,13]. A variety of ultrasound-guided local nerve blocks have recently been described in rabbits [14,15].

Both an ultrasound-guided approach to the brachial plexus block and a combined ultrasound and peripheral nerve stimulator-guided approach have been described in rabbits [16,17], both of which concluded that the techniques are feasible, reproducible, and safe, providing adequate analgesia for rabbit thoracic limb surgery.

The aim of this study was to assess the efficacy of a solely nerve stimulator-guided axillary approach to the brachial plexus block for rabbits undergoing thoracic limb orthopaedic surgery. In this study, the brachial plexus block with lidocaine was compared to a lidocaine constant rate infusion (CRI), which has previously been demonstrated to maintain gastrointestinal motility and provide analgesia for soft tissue surgical procedures [18].

We hypothesised that the nerve stimulator-guided brachial plexus block would provide effective intraoperative analgesia. In addition, we hypothesised that an intraoperative lidocaine CRI would increase faecal output postoperatively.

## 2. Materials and Methods

This prospective randomised study was conducted as a spin-off study of a larger project assessing the efficacy of bone scaffolds for correcting critical-size bone defects in rabbits. For the current study, 19 rabbits undergoing experimental thoracic limb surgery in general anaesthesia with isoflurane were either subjected to a brachial plexus block with lidocaine (group BPB) or to an intraoperative lidocaine CRI (group CRI). Rabbits were evenly allocated to either group via lot prior to the surgery. Perioperative rescue analgesia and postoperative food intake, faecal output, and behaviour were assessed at 30, 60, 90, and 120 min, and 10 and 18 h post-recovery, and a compared between the groups.

Experimental protocols were approved by the United Kingdom Home Office as governed by UK law under the Animals (Scientific Procedures) Act 1986, project license number PP1153947, and abide by the ARRIVE guidelines. The work was conducted at the University of Cambridge Biomedical Services research animal facilities. A total of 19 female New Zealand white rabbits (15 purchased from ENVIGO RMS LTD-Loughborough, UK; 4 from Charles River UK Limited-Margate, UK), aged 12 weeks, were acclimatised to their housing for 4–6 weeks prior to starting the study.

### 2.1. Animals and Housing

The rabbits were housed in pairs in floor pens with overall dimensions of 150 cm width, 150 cm depth, and no ceiling. Each cage contained an elevated resting platform 25 cm high, which also served as a shelter. After surgery, the rabbits were housed individually, with the original floor pens divided in half, each containing an elevated resting platform. They continued to have visual and tactile contact with their previous companion. All animals had ad libitum access to food and water during the experimental period. The food consisted of hay and a commercial dry pellet food formulated for rabbits. As environmental enrichment, a selection of dried herbs and vegetables or fresh carrots was supplied once daily, though these were withheld for 24 h following surgery. Each pen also contained a large cardboard tube, which provided environmental enrichment and shelter. Water was supplied in water bottles, which were refreshed once a day.

### 2.2. Anaesthesia and Monitoring

Prior to each surgery, a clinical examination of each animal was performed. Premedication consisted of medetomidine 100 µg kg^−1^ (Sedator, Dechra, Northwich, UK), fentanyl 5 µg kg^−1^ (Fentadon, Dechra, Northwich, UK), and midazolam 0.5 mg kg^−1^ (Hypnovel, Accord-UK Ltd., Middlesex, UK) administered intramuscularly. After the onset of sedation (15 min later), an intravenous (IV) cannula was placed in the left marginal auricular vein, and anaesthesia was induced with propofol (Propoflo Plus, Zoetis, Leatherhead, UK), 1–3 mg kg^−1^, administered to effect. The rabbits were intubated using a capnograph-guided technique. If there were three unsuccessful attempts to intubate, a supraglottic airway device (V-gel^®^, Docsinnovent, Hemel Hempstead, UK) was placed. The airway device was connected via a heat and moisture exchanger (HME) (Clear-Therm™ Micro HMEF, Intersurgical, Wokingham, UK) non-rebreathing system. Anaesthesia was maintained with isoflurane (Isofane, Covetrus, Dumfries, UK) to maintain a sufficient anaesthetic level to facilitate the procedure on 100% oxygen. All anaesthesia was performed by two experienced anaesthetists (SM and CG), and adjustments to the level of anaesthesia were based on reaction to surgical stimulus: increased heart rate (HR) or respiratory rate (*f*R), breathing against a ventilator, or movement. Routine intraoperative monitoring included sidestream capnography, fraction expired inhalant (FE’ISO%), oesophageal temperature, pulse oximetry, and oscillometric blood pressure with a Mindray Beneview T8 multiparameter monitor. Physiological parameters were monitored continuously and recorded every five minutes. Recordings of HR, assessed by pulse oximetry and manual pulse palpation, were averaged in each rabbit for “HR pre-op” and “HR intra-op”. The HR change was calculated based on the difference between these two periods.

Intermittent positive pressure ventilation (IPPV) was used to maintain end-tidal carbon dioxide partial pressure at 35–50 mmHg using a mechanical thumb ventilator. Hypotension (MAP < 65 mmHg) or bradycardia (HR < 140 bpm) were treated at the discretion of the anaesthetist. Lactated Ringers solution (Aqupharm 11, Animalcare, York, UK) was administered at 10 mL kg^−1^ h^−1^ during the anaesthesia. No intravenous fluids were administered postoperatively. An electric heat mat was used to maintain normothermia (38–39.9 °C) throughout the anaesthesia.

The left thoracic limb was clipped and aseptically prepared for radial ostectomy according to the guidelines of the main orthopaedic study. After aseptic preparation of the limb, the rabbits were administered either a brachial plexus block with lidocaine (Hameln Pharma, Gloucester, UK) (group BPB) or an IV bolus of lidocaine followed by a CRI (group CRI).

Rabbits were administered rescue analgesia (fentanyl 5 µg kg^−1^) if the HR or fR increased by 20% or more from the individual baseline during the surgery. Baseline values were recorded immediately prior to the surgical start time. Rabbits receiving rescue analgesia remained within the study. Carprofen 4 mg kg^−1^ (Rimadyl, Pfizer, Tadworth, UK) subcutaneously (SC) was administered at the end of surgery and once every 24 h for five days thereafter.

### 2.3. Brachial Plexus Block

Ten rabbits received a peripheral nerve stimulator-guided brachial plexus block. After intubation, rabbits were placed in right lateral recumbency, allowing access to the left thoracic limb. The cranial aspect of the shoulder and ventral neck were clipped and aseptically prepared. The positive electrode of the nerve stimulator was positioned on the lateral aspect of the left elbow. The landmarks used were the acromion, the cranial border of the greater tubercle of the humerus, and the cranial border of the first rib. The insulated needle was inserted cranially to the acromion, immediately dorsal to the clavicle, and advanced in a ventral and caudal direction, parallel to the longitudinal access of the vertebral column and thoracic wall. The needle was advanced slowly with an initial current of 2 mA and monitored for nerve stimulation corresponding to radial nerve stimulation: extension of the elbow, carpus, and digits. Once the appropriate response had been elicited, the current was reduced incrementally to 0.5 mA to ensure proximity to the nerve. Once the position was confirmed, the current was reduced to 0.2 mA to rule out the intraneural placement of the needle tip. Intravascular placement was excluded by aspiration prior to 0.1 mL kg^−1^ of lidocaine 2%. The needle was then withdrawn approximately 0.5 cm, and a further 0.1 mL kg^−1^ of lidocaine 2% was injected following aspiration. This was repeated once more to give a total volume of 0.3 mL kg^−1^ of lidocaine 2%, equating to a total dose of 6 mg kg^−1^.

### 2.4. Lidocaine CRI

Eight rabbits received lidocaine 2 mg kg^−1^ IV over 5 min during surgical preparation of the limb, followed by an infusion of 100 µg kg^−1^ min^−1^ for the duration of the surgical procedure. The infusion was delivered using a calibrated syringe driver (BD Alaris Syringe Pump, BD, Wokingham, UK) and was stopped at the same time as the isoflurane.

### 2.5. Surgery

All rabbits underwent the same surgical procedure. During surgery, the rabbits were positioned in left lateral recumbency to allow a medial approach to the left antebrachium. A titanium K wire was positioned in the radius in a mediolateral orientation as a marker for postoperative radiographic analysis. A 15 mm full-thickness defect was created in the radius diaphysis by two osteotomies and then the cutting of the interosseus ligament between the radius and the ulna. The defect was then filled with bone scaffold as part of the primary orthopaedic study. No further stabilisation of the bone was provided. The surgical site was then closed, and radiographs were acquired prior to recovery.

### 2.6. Recovery and Postoperative Assessment

Rabbits were extubated when a swallow reflex was present and were returned to individual enclosures with clean bedding once they were able to support their heads and maintain sternal recumbency. The time from turning the isoflurane off to extubation was recorded as the recovery time.

The rabbits were filmed in their enclosures for the first two hours immediately following their return using a GoPro camera (Hero 4, GoPro, London, UK). This was fixed at the height of 140 cm above the ground to the centre of one end of the enclosure, angled to include the entire enclosure. The camera position was consistent for each rabbit to allow for post hoc video processing.

The rabbits were pain scored by either an appointed observer (SM or CG) or both if observations were during daytime working hours, using the Rabbit Grimace Scale and the Bristol Rabbit Composite Pain Scale 30, 60, 90, and 120 min, and 10 and 18 h post-recovery. If both observers were present, the scores were discussed to try and maintain consistency between observers when scoring alone. During the first 90 min, buprenorphine 0.05 mg kg^−1^ SC (Buprecare, Animalcare, York, UK) was administered as rescue analgesia if the RGS score was greater than 5/10. If no buprenorphine was administered beforehand, one dose was given at T120 to ensure adequate pain relief following the orthopaedic surgery. In all rabbits, another dose of buprenorphine (0.05 mg kg^−1^ SC) was administered at T10h and T18h. The time points were chosen to coincide with the expected duration of analgesia provided by buprenorphine, which was approximately eight hours [19]. At T10h and T18h, pain scores were recorded prior to the administration of buprenorphine.

To assess food intake, the food pellets were weighed upon the return of the rabbits to their enclosures and then weighed at the above time points. Rabbits had continuous access to hay as it was fed loose, but the intake amount could not be assessed. Instead, hay consumption was classified as “eating”, “showing interest in food”, or “hay undisturbed” based on the animal behaviour at the predetermined observation points (30, 60, 90, and 120 min, and 10 and 18 h post-surgery). As a substitute for gastrointestinal motility, faecal output was assessed by collecting the faeces produced by each rabbit at each time point. The morphology was subjectively assessed each time as “normal” or “abnormal”. “Normal” faeces were defined as uniform, round, and smooth. “Abnormal” faeces were irregular in size and shape, sticky, or crumbly. The rabbits were weighed daily for three days after surgery.

### 2.7. Post Hoc Video Processing

Videos obtained from each rabbit during the immediate recovery period (up to T120) were automatically processed afterwards. For this purpose, a computer programme was designed using the Python (Python Software Foundation, version 3.10.5) coding language to detect the whole body movement of the rabbits (Appendix A), such as hopping or crawling. It automatically analysed each video and was designed to detect the movement of the white rabbit against the contrasting background, allowing automatic measurement of the distance travelled by each rabbit during the first two hours after recovery. Random clips from the videos were used to refine the limits of the movement detector prior to video processing and ensure that only whole rabbit movement was detected, excluding partial movement such as head turning. The video processing was observed as further confirmation that rabbit movement was detected correctly. The videos were also used to manually assess the presence or absence of the following behaviours in the first two hours postoperatively in each rabbit: eating hay or pellets, generalised grooming, grooming the affected limb, hopping, sprawling, and interacting with their environment. Interacting with their environment included hopping through or chewing their cardboard tube or interacting with their bedding material.

### 2.8. Statistics

Data were assessed for normality by inspection of QQ plots. Bartlett’s and Levene’s tests were used to assess the equality of variance. Data were reported as median (first quartile, third quartile) when not normally distributed or as mean (standard deviation) when normally distributed. The Mann–Whitney U test and descriptive statistics were used to compare demographic and surgical data. Fisher’s exact test was used to compare the requirement for rescue analgesia intraoperatively and postoperatively and to compare the number of rabbits who produced faces in the first two hours postoperatively. Pain scores between groups BPB and CRI were examined using descriptive statistics. Data were compared using commercially available software (R: R Studio, version 4.2.1; JASP: version 0.17.3), and significance was interpreted at *p* < 0.05 where applicable.

## 3. Results

Of the 19 initially included rabbits, one rabbit (allocated to group CRI) had to be excluded from the study due to persistent preoperative tachypnoea. From the remaining 18 rabbits, which initially entered the study, one animal (allocated to group BPB) was retrospectively excluded due to surgical complications. The remaining 17 rabbits completed the surgery and had an uneventful recovery, with nine rabbits in group BPB and eight animals in group CRI for data analysis. There was no difference in preoperative rabbit weight between the groups (Table 1). The general anaesthesia time was 115 ± 14.1 min in group BPB and 117 ± 20.9 min in group CRI, showing that the brachial plexus block did not significantly increase total anaesthesia time (*p* = 0.63). There were no differences in surgery or recovery time (Table 2), and all rabbits were returned to their pens within 15 min of extubation.

The total dose of propofol required for intubation was similar in all rabbits, with 3.55 ± 1.5 mg kg^−1^ and 2.56 ± 2.21 mg kg^−1^ in groups BPB and CRI, respectively (Table 2). All of the rabbits in the CRI group were intubated successfully with a 3.5 mm internal diameter PVC endotracheal tube, whereas 5/9 in the BPB group were intubated with supraglottic airway devices placed in the remaining four rabbits. In the latter cases, laryngeal masks were placed after three unsuccessful attempts to orotracheally intubate.

Mean FEISO (%) during surgical stimulation was significantly lower in group BPB than in group CRI (Table 2). Baseline values for HR obtained prior to the start of surgical stimulation were significantly lower in group BPB compared to group CRI (Table 2). In addition, the intraoperative HR was lower in group BPB than in group CRI. However, the change in HR between preoperative and intraoperative groups was not significantly different between the groups.

The oscillometric blood pressure proved to be unreliable, providing inconsistent readings at multiple time points in many of the rabbits. Hence, these data were excluded from any analysis.

All of the rabbits in group CRI required at least one bolus (range: 1–3 boluses) of rescue analgesia during surgery, whereas no rabbit in group BPB required rescue analgesia (Table 3; *p* < 0.01). However, in the first two h postoperatively, there was no difference in the number of rabbits in each group receiving rescue analgesia (Table 3; *p* = 0.49).

All rabbits had a temperature of at least 37 °C at the time of recovery, measured by oesophageal and rectal thermometers.

During the postoperative observation period, faecal production was similar between group BPB and group CRI. Within 120 min, six out of nine rabbits in group BPB and four out of eight rabbits in group CRI produced faeces with normal morphology, with the remaining rabbits producing faeces of normal morphology within the first ten h (Table 4). All rabbits were observed eating hay or pellets during the first two h.

Postoperative behavioural assessment showed a similar occurrence of grooming of the operated front leg during the first 2 h (Table 4). All rabbits showed hopping, grooming themselves in a generalised manner, and interacting with their environment (Table 4).

Post hoc analysis showed a similar activity budget for the first 2 h with similar distances travelled in group BPB (mean = 17.4 ± 10.77 m) and group CRI (mean = 16.14 ± 7.97 m; *p* = 0.89).

The postoperative pain scores are reported in Table 5 and Table 6. The pain scores using both scales decreased in value with increasing time after surgery.

Three days postoperatively, all rabbits except one had lost weight. There was no significant difference in weight loss between the groups (Table 1), and no weight loss was of clinical concern.

## 4. Discussion

In this prospective study, the peripheral nerve stimulator-guided brachial plexus block with lidocaine provided sufficient intraoperative analgesia, but the lidocaine CRI did not, as evidenced by the unanimous requirement for rescue analgesia in the CRI group. Our hypothesis that the nerve stimulator-guided brachial plexus block would provide effective intraoperative analgesia was thus confirmed. In addition, we hypothesised that an intraoperative lidocaine CRI would increase faecal output postoperatively, but this was not observed, with no difference in food intake or faecal output between the groups observed.

The effectiveness of the brachial plexus block is in line with studies investigating the efficacy of brachial plexus blocks [16,20,21]. A successful block was assumed in all cases by a consistent lack of response to surgical stimulation. All BPB rabbits exhibited incomplete motor function of the thoracic limb during the first two hours of recovery, but it was difficult to assess the return of normal motor function due to the nervous temperaments of the rabbits and the lameness associated with the surgical procedure. Potential complications associated with brachial plexus blocks include intraneural or intravascular injection, pneumothorax, and, rarely, hemi-diaphragmatic paralysis following phrenic nerve anaesthesia. There were no complications associated with the brachial plexus block noted in this study. Two rabbits, one from each group, were euthanised seven days after surgery due to necrosis of the paw of the left thoracic limb, immediately distal to the surgical site. This was assumed to be unrelated to the brachial plexus block since only one of the rabbits was in group BPB, and the region of necrosis was in direct proximity to the surgical site.

There are two previous studies describing brachial plexus blockade in rabbits, one using an ultrasound-guided axillary approach [16] and the other describing a combined ultrasound and nerve stimulator-guided axillary approach [17]. One reported advantage of using ultrasound guidance is the smaller volume of local anaesthetic required to achieve an effective nerve block. However, the total volume used in this study, 0.3 mL kg^−1^, was smaller (0.7–0.8 mL kg^−1^ [16]) or comparable (0.8 ± 0.3 mL in rabbits with a mean weight of 2.5 kg [17]) to previous studies. In this study, adequate coverage of all nerves within the brachial plexus was not visually confirmed, but the efficacy of the block was assumed in all cases by a lack of expected response to surgical stimulus and no requirement for intraoperative rescue analgesia.

Although lidocaine CRIs have previously been found effective in managing pain associated with soft tissue surgical pain in rabbits [18], it was not sufficient as a sole analgesic technique for orthopaedic surgery in this study, which could be due to the more severe pain associated with orthopaedic surgeries. Intravenous lidocaine is associated with reduced postoperative pain and decreased recovery times in humans [22] and dogs [23] and has dose-dependent minimum alveolar concentration (MAC) sparing effects in many species. In rabbits [24], lidocaine infusions of 50 µg kg^−1^ min^−1^ and 100 µg kg^−1^ min^−1^ reduced the MAC of isoflurane by 12% and 21.7%, respectively. In contrast, in the current study, the mean end-tidal isoflurane concentration required to maintain stable anaesthesia was lower in group BPB, which could be due to insufficient analgesia in group CRI. Lidocaine overdose can result in adverse effects such as tremors, convulsions, and arrhythmias, including bradycardia and prolonged PR and QRS intervals [25,26]. Rabbits have a high LD50 of 20 mg kg^−1^ [26], making lidocaine infusions relatively safe compared to other domestic species, and no adverse effects were observed during this study. The short duration of infusion during this study reduced the accumulation of the drug, which reduced the risk of adverse effects despite the high dose and loading dose used. The same infusion rate of 100 µg kg^−1^ min^−1^ has been used for two days without complications observed in rabbits following ovariohysterectomy [18].

The intraoperative differences in rescue analgesia were not reflected in the postoperative period in this study, during which there were no differences in pain scores or requirements for rescue analgesia. This could be due to the limited duration of action of lidocaine block in group BPB of 1–2 h into the postoperative period. However, only five out of 17 rabbits required rescue analgesia in the two hours immediately post-op. One reason for this may have been the limited ability to accurately detect pain. Due to the difficulty of assessing pain in prey species such as rabbits, multiple methods were used in this study. Both the Rabbit Grimace Scale [27] and the Bristol Rabbit Pain Scale [28] were used. The Rabbit Grimace Scale uses facial expressions to evaluate pain. Based on the mouse and rat grimace scales [29,30], it uses similar facial action units, which include orbital tightening, cheek flattening, nostril shape, whisker shape and position, and ear shape and position. It was found to be more reliable than behavioural markers of pain in rabbits undergoing ear tattooing. There are no pain scales validated for orthopaedic pain in rabbits. The Bristol Rabbit Pain Scale is a composite pain scale designed to aid pain assessment in rabbits experiencing acute pain. Since data collection, this scale has been validated for acute pain associated with ovariohysterectomy and orchiectomy [31]. This scale was used as a secondary indicator of pain after the Rabbit Grimace Scale to improve the ability to detect pain following surgery in these rabbits. The pain scales exhibited a progressively stronger correlation with time. Neither scale is appropriate for use in sedated patients, and the scores in the first two hours following surgery may have been affected by residual sedation. All of the rabbits were observed hopping, eating, and interacting with their environment during this time, but it is impossible to exclude the effect of anaesthesia on the pain scores during this period.

Food intake, faecal output, and weight loss were assessed in this study because changes in both parameters have been associated with pain in laboratory animals [32,33,34]. Both pain and a change in food intake are risk factors for gastrointestinal stasis and are indicated by a reduction in food intake and a change in the number and morphology of faecal pellets. No control observations were made prior to surgical intervention, but the faecal output was subjectively reduced compared to the author’s experience of normal faecal output in rabbits. However, none required intervention for gastrointestinal stasis or the management of excessive weight loss. A single dose of buprenorphine (100 µg kg^−1^ IM) [35] does not appear to reduce gastrointestinal motility in rabbits that have not been anaesthetised, but general anaesthesia followed by buprenorphine (30 µg kg^−1^) TID did increase gastrointestinal transit time and reduce faecal output [36]. Any prokinetic effects of the intraoperative lidocaine administration may have been reduced by the postoperative administration of buprenorphine.

The rabbits’ activity and behaviour were also monitored to gain a thorough assessment of comfort and welfare. Ethograms, nesting behaviour, and burrowing behaviour have all been used extensively to assess pain in laboratory rodents [37,38] but have been the subject of very limited investigation in rabbits. The object tracker reliably detected the movement of the rabbits after the parameters were manually adjusted to filter out isolated head movement. Movement parameters have been included in some behaviour-based pain scales, but none are validated. This study found a very large variation in the distance travelled by the rabbits in each group, with no significant difference between groups, suggesting that it is not a useful indicator of pain. However, the postoperative pain scores were also similar between groups, so the value of this information in this study is limited. Pain may be expressed through general behaviour changes [32] or specific changes in response to the painful region. In this study, twice as many rabbits in group BPB were observed grooming the affected leg during the first two hours as in group CRI, which was unexpected. The return of normal sensation in the blocked limb may cause discomfort that causes the rabbits to groom more than those who retained normal sensation throughout. Though this difference was statistically insignificant, this study was underpowered in this regard, and specific behavioural changes in response to pain and regional anaesthesia warrant further investigation in rabbits.

In an attempt to monitor pain as thoroughly as possible, both direct observation and videography were used to assess different parameters. Distant monitoring via video allows us to better assess natural behaviour and increases the possibility of objective measures such as distance travelled and duration spent performing behaviours of interest. However, direct observation allows for a closer examination of the demeanour, alertness, and facial signs of pain. It also allows for quantitative measurement of food intake and faecal output. A combined approach may provide the most effective method of assessing pain in rabbits who are less accustomed to human contact.

There were a number of limitations in this study. The sample size was controlled by a primary orthopaedic study and limited the power to detect differences in postoperative pain score and behaviour between the groups, such as grooming the affected leg. Only half of the rabbits in group BPB were not tracheally intubated but had supraglottic airway devices. This may have changed the vagal stimulation experienced by group BPB compared to group CRI and caused changes to parameters such as heart rate that could not be quantified. The observers were not blinded to the groups, which may have introduced bias into the pain scores. There were two observers conducting postoperative assessments, which ultimately might have affected the results. During the working day, the observers were both present and discussed the use of the pain score scales in an attempt to maintain consistency, but it was not possible to formally assess this during this study. As a result, the impact of having more than one observer is unknown but is believed to be small. Many of the other parameters measured were objective measures recorded as either ‘present’ or ‘absent’ or were measured by the computer programme, which reduces the potential to introduce bias to the results. The rabbits were acclimatised to receiving loose hay, which was difficult to weigh accurately. Feeding only pellets for the duration of the study would have allowed for accurate measurement of food intake, but this diet change may have increased the risk of gastrointestinal disturbance and ileus. As a result, we simply recorded whether the rabbits had eaten or not, but food intake could be better quantified in future studies. After the first two hours, the frequency of pain scoring decreased and limited the observer’s ability to detect differences between the groups. Between the study timepoints, the rabbits were observed by trained staff at the facility. The study team was notified if any individuals were exhibiting abnormal behaviour or signs of pain, but these were not included in the study in order to reduce the number of observers and inter-rater variation. No concerns were raised between the study time points. Additionally, the rabbits were separated after the surgery to minimise activity and trauma to the surgical site caused by their companions. They were still able to see and touch their previous companion through the pen bars, but the separation may have increased stress. This, in turn, may have influenced behaviour, food intake, and face production. No concerns were raised by the observers or facility staff, but as prey animals, signs of stress may be subtle and easily missed.

## 5. Conclusions

In conclusion, the axillary approach to the brachial plexus block, guided by a peripheral nerve stimulator, was straightforward to administer and effective in providing analgesia for thoracic limb surgery in rabbits. No complications attributable to the block were reported in this study.

## Figures and Tables

**Table 1 vetsci-11-00213-t001:** The bodyweight of the rabbits receiving either a brachial plexus block (BPB, *n* = 9) or a lidocaine CRI (CRI, *n* = 8) preoperatively and the bodyweight change three days postoperatively. Data for weight (kg) are given as median (IQR), and data for weight change are given as mean (SD).

Variable	Group		Significance
	BPB	CRI	
Weight (kg)	3.4 (3.22–3.62)	3.4 (3.35–3.62)	*p* = 0.63
Weight Change 3 Days Post-surgery	−0.075 (0.046)	−0.091 (0.042)	*p* = 0.18

**Table 2 vetsci-11-00213-t002:** Descriptive data of anaesthesia monitoring in rabbits undergoing experimental thoracic limb surgery with either brachial plexus block (BPB, *n* = 9) or intraoperative lidocaine (CRI, *n* = 8). Data are given as the mean (SD).

Variable		Group		Significance
		BPB	CRI	
General anaesthesia time (minutes)		115 (14.1)	117 (20.9)	*p* = 0.63
Surgical time (minutes)		51 (6.4)	53 (10.5)	*p* = 0.84
Recovery time (minutes)		9.9 (4.1)	10.3 (3.9)	*p* = 0.81
Induction dose of propofol (mg kg^−1^)		3.55 (1.5)	2.56 (2.21)	*p* = 0.28
FE’ISO (%)		1.51 (0.17)	1.71 (0.04)	*p* = 0.01 *
Heart rate (bpm)	Pre-op	175.2 (16.6)	190.2 (11.4)	*p* = 0.04 *
	Intra-op	163.7 (16.2)	190.4 (19.9)	*p* = 0.01 *
	Differences between pre-op and intra-op	−11.5 (3.8)	0.2 (4.0)	*p* = 0.18

* Significant result: *p* < 0.05; FE’ISO (%) = fraction of expired isoflurane (%); bpm = beats per minute.

**Table 3 vetsci-11-00213-t003:** The frequency of intraoperative and postoperative rescue analgesia administration in group BPB and group CRI. Fisher exact tests were used to compare the frequencies of rescue analgesia administration.

Rescue Analgesia	Group		Significance
	BPB	CRI	
Intraoperative administration	0/9	8/8	*p* < 0.01 *
Postoperative administration	2/9	3/8	*p* = 0.49

* Significant result; *p* < 0.05.

**Table 4 vetsci-11-00213-t004:** The frequency of eating, normal faeces production, and recorded behaviours in groups BPB and CRI. Fisher exact tests were used to compare the frequencies of the behaviours in each group.

Behaviour	Group		Significance
	BPB	CRI	
Normal faeces in the first two h	6/9	4/8	*p* = 0.63
Eating hay and/or pellets	9/9	8/8	*p* = 1
Grooming the Affected Limb	8/9	4/8	*p* = 0.13
Generalised grooming	9/9	8/8	*p* = 1
Hopping	9/9	8/8	*p* = 1
Sprawling	6/9	4/8	*p* = 0.63
Interacting with Environment	9/9	8/8	*p* = 1

**Table 5 vetsci-11-00213-t005:** Postoperative pain scores using the Rabbit Grimace Scale in rabbits undergoing experimental thoracic limb surgery with either brachial plexus block (BPB, *n* = 9) or intraoperative lidocaine (CRI, *n* = 8). Data are expressed as median (IQR).

Timepoint	Group	
	BPB	CRI
T0	4 (3–4)	4 (4–4)
T30	4 (3–4)	4 (3–4)
T60	3 (2–4)	4 (3–5)
T90	3 (2–4)	4 (4–6)
T120	3 (2–4)	4 (3–5)
T10h	2 (1–3)	2 (1–3)
T18h	2 (1–3)	2 (1–4)

**Table 6 vetsci-11-00213-t006:** Postoperative pain scores using the Bristol Rabbit Pain Scale in rabbits undergoing experimental thoracic limb surgery with either brachial plexus block (BPB, *n* = 9) or intraoperative lidocaine (CRI, *n* = 8). Data are expressed as the median (IQR).

Timepoint	Group	
	BPB	CRI
T0	8 (6–10)	10 (8–11)
T30	9 (7–11)	10 (8–10)
T60	6 (5–10)	9 (8–10)
T90	7 (4–9)	8 (7–10)
T120	5 (4–8)	8 (6–9)
T10h	4 (3–6)	6 (3–7)
T18h	3 (2–5)	3 (2–4)

## Data Availability

The data presented in this study are available upon request from the corresponding author (accurately indicating status).

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
