# Peer review of "Evaluating the Efficacy of a Peripheral Nerve Simulator-Guided Brachial Plexus Block in Rabbits Undergoing Orthopaedic Surgery Compared to Systemic Analgesia"

_vetsci, 2024, doi:10.3390/vetsci11050213_

Round 1

Reviewer 1 Report

Comments and Suggestions for Authors

It is potentially very interesting article. The comparison of systemic and local anesthesia techniques is currently interesting in anesthesia research. The rabbit is really nice as animal model. It is unknown a lot of things in rabbit anesthesia. The authors described all procedures in details.

I recommend several corrections.

1. One rabbit was excluded, in Materials and methods due to tachypnea, in Results due to tachycardia. Please correct. It is not necessary to repeat cause of exclusion in Materials and methods and Results.

2. Line 134- . All anaesthetics were performed - Do you mean all anesthesia?

3. Somewhere for drugs and equipment use name of drug, manufacturer and country, somewhere without country, somewhere only name of drug or equipment. Please correct.

4. I hope that in all rabbits same orthopedic surgery was performed. Can you declare it?

Reviewer 2 Report

Comments and Suggestions for Authors

Dear Authors, congratulations on your great work. I think the paper is well studied, explained, and discussed. However, I have some minor points that could be revised for further improvement. Please find my comments enclosed below:

Line 91: Why 19 rabbits? How did you arrive at this number?

Lines 95-96: I believe it would be beneficial if you briefly explain why, you chose these specific times, especially T10h and T18h.

Line 214: Were the evaluations of all rabbits conducted by the same observers? How many observers were involved in the scoring process? If there were multiple observers, is there a possibility that this might have influenced the ultimate score? This aspect could be further elaborated upon in the discussion section.

Lines 237-239: Has this program been utilized previously? How can you ensure that this system functions accurately and validate the exported results?

Line 272: Please include the p-value, when referring to significance or lack thereof in the text.

Table 1: Are the values presented for weight in this table indicative of the mean (minimum-maximum) or the median (interquartile range)?

I presume that the weight values presented here are reported as the median (IQR), as specified in the materials and methods section. However, the table caption states that the data are presented as the mean (SD), which is correct for Weight change but not weight.

Table 2 and 3, Significance: Please decide whether to use values up to two decimals or three decimals consistently.

Table 2, FE’ISO(%): Please explain abbreviations in the table footer when they're first mentioned in the table.

Table 2, Heart rate: please add unit.

Table 2, change: Is it the intra-operative change of HR or changes from the beginning of anesthesia? It was clarified in lines of 140-143. Here, you could provide additional information to enhance clarity. For instance:

Heart rate: surgical preparation or pre-op, intra-op, difference between pre- and intra-op

Line 295: Please attempt to maintain the same order of parameters in the text as they appear in the table, as closely as possible. For instance, if „FE'ISO (%)" precedes "Heart rate" in the table, discuss "FE'ISO (%)" before addressing "Heart rate" in the text.

Line 296: Do you mean Table 2?

Lines 305-306: You do not need to repeat the results in the caption of the table, as it was mentioned in the text.

Lines 329-330: Was the difference between groups in pain scores significant? For instance, in Table 5, T90: 3(2-4) vs. 4 (4-6)? I think, instead of stating "a similar trend " a concise statement would clarify the findings.

Line 347: After providing a summary of the main results at the beginning of the discussion, please mention the hypotheses you formulated and whether these hypotheses were confirmed or not.

Line 373-374: If there is any source or reference available for this information (… not sufficient for Ortho…), please include the citation here.

Line 451: My inquiry about the calculation of sample size has been addressed here.

References: References need to be reviewed regarding formatting, especially at the beginning of reference Nr. 9 and Nr. 14-39, are some numbers, probably from the old format of the paper that should be removed.

Reviewer 3 Report

Comments and Suggestions for Authors

79 - The goals of this study are a little muddy. You state that there has not been a solely NS guided BPB in rabbits and the goal of your study is to "assess the efficacy of a solely nerve stimulator guided axillary approach to the brachial plexus block for rabbits".  Yet the title states you are comparing the efficacy of a BPB to a lidocaine CRI.  Are you trying to assess which analgesic technique is better or whether a solely NS guided BPB is effective?

Then as additional goals you add in GI motility but state that one should not affect it and one should increase it..which are two separate goals. 

123 & 265 - one states a rabbit was excluded due to tachypnea and one says due to tachycardia.  Were there two rabbits excluded?

360 - how do you know it is unrelated to the block?

Limitations:

Separating the rabbits after surgery but not before may create stress unrelated to the study. 

Almost half of the BPB group were not tracheally intubated which may cause a different amount of vagal stimulation 

Conclusion:  This needs to focus on the main goal of the study which is still a little muddy.  The addition of pains scales is another goal. 
